# Multicentre Evaluation of the EUCAST Rapid Antimicrobial Susceptibility Testing (RAST) Extending Analysis to 16–20 Hours Reading Time

**DOI:** 10.3390/antibiotics11101404

**Published:** 2022-10-13

**Authors:** Gabriele Bianco, Donatella Lombardo, Guido Ricciardelli, Matteo Boattini, Sara Comini, Rossana Cavallo, Cristina Costa, Simone Ambretti

**Affiliations:** 1Microbiology and Virology Unit, University Hospital Città della Salute e della Scienza di Torino, 10126 Turin, Italy; 2Operative Unit of Microbiology, IRCCS Azienda Ospedaliero-Universitaria di Bologna, 40138 Bologna, Italy; 3Department of Public Health and Paediatrics, University of Torino, 10126 Turin, Italy

**Keywords:** EUCAST RAST, blood culture, rapid susceptibility testing, ceftazidime/avibactam, ceftolozane/tazobactam

## Abstract

The aim of the study was to evaluate the EUCAST RAST method by extending analysis to 16–20 h reading time and performance with new β-lactam/β-lactamase inhibitor combinations. A total of 676 positive blood cultures (BCs) were enrolled. Results at 4 h, 6 h, 8 h and 16–20 h were interpreted according to bacterial species using EUCAST RAST breakpoints (version 5.1). For species for which no breakpoints were available, tentative breakpoints were used. Categorical agreement with the Microscan microdilution system was analysed. Among the 676 BCs enrolled, 641 were monomicrobial and were included in the analysis. Categorical agreement ranged from 98.9% at 4 h to 99.4% at 16–20 h. The rates of very major errors were 3.3%, 3.7% and 3.4% at 4 h, 6 h and 8 h, respectively, and decreased to 1% at 16–20 h (*p* < 0.001). The number of major errors was low for each reading time (0.2% and 0.4% at 4 h and 6 h, respectively, and 0.3% at both 8 h and 16–20 h). The proportions of results in the area of technical uncertainty were 9.9%, 5.9%, 5% and 5.2% for readings at 4 h, 6 h, 8 h and 16–20 h, respectively. Tentative breakpoints proposed for Enterobacterales other than *E.coli*/*K.pneumoniae* and coagulase-negative staphylococci showed overall performances comparable to those observed for *E. coli*/*K. pneumoniae* and *S. aureus.* In conclusion, EUCAST RAST has been shown to be reliable to determine microbial susceptibility to main antimicrobials, including ceftazidime/avibactam and ceftolozane/tazobactam. A poorer performance was observed for certain species/antimicrobial agent combinations. The better performance observed at 16–20 h compared to the early readings may confer to the method greater potential for antimicrobial de-escalation interventions.

## 1. Introduction

Bloodstream infections (BSIs) are a major cause of morbidity and mortality with increasing incidence worldwide, often due to delayed or inappropriate antimicrobial treatment [1,2,3]. Hence, blood cultures (BCs) remain essential to identifying causative pathogens and their antimicrobial susceptibility profiles. Several molecular-based systems and MALDI-TOF MS approaches have been implemented directly from positive BCs [4,5]. However, given the wide range of antibiotic-resistance mechanisms and their increasing spread worldwide, rapid antimicrobial susceptibility tests (RASTs) have shown potential for shortening time to effective treatment and consequently improving clinical outcomes in BSIs [6,7,8]. Recently, EUCAST developed a phenotypic RAST method based on disc diffusion and performed directly from positive BC bottles to provide results after 4 h, 6 h and 8 h of incubation [9,10,11]. The method has currently been validated for four Gram-negative species (*Escherichia coli*, *Klebsiella pneumoniae*, *Pseudomonas aeruginosa* and *Acinetobacter baumannii*) and four Gram-positive species (*Staphylococcus aureus*, *Enterococcus faecalis*, *Enterococcus faecium* and *Streptococcus pneumoniae*) [12]. However, tentative RAST breakpoints for related bacterial species (e.g., Enterobacterales species other than *E.coli*/*K.pneumoniae* (other-EB) and coagulase-negative staphylococci) have been proposed and positively evaluated [13]. Additionally, EUCAST RAST has been recently validated for 16–20 h of incubation, with specific reading guidelines and interpretive criteria for situations in which readings after shorter incubation/reading times cannot be performed due to lab-logistic reasons [12].

Despite several reports have shown that EUCAST RAST represents a reliable method for rapid determination of *in vitro* susceptibility towards several antimicrobials [10,11,14,15], data on its performance for new β-lactams/β-lactamase inhibitors (e.g., ceftazidime/avibactam and ceftolozane/tazobactam) are lacking [16]. To date, only one study has evaluated EUCAST RAST on bacterial species for which specific RAST breakpoints are not available [13], and no data are yet available on the performance of EUCAST RAST with readings at 16–20 h.

The aim of this study was to evaluate the EUCAST RAST method extending the analysis to 16–20 h reading time, new β-lactam/β-lactamase inhibitor combinations and performance for other-EB and coagulase-negative staphylococci.

## 2. Results

Among the 676 BCs enrolled, 641 were monomicrobial and were included in the analysis. Bacterial isolates included *E. coli* (*n* = 191), *K. pneumoniae* (*n* = 87), *P. aeruginosa* (*n* = 42), *A. baumannii* (*n* = 15), other-EB (*n* = 61), *S. aureus* (*n* = 98), coagulase-negative staphylococci (CoNS) (*n* = 72), *E. faecalis* (*n* = 48) and *E. faecium* (*n* = 27). Overall, there were 4058 antibiotic/bacteria combinations (*E. coli, n* = 1470; *K. pneumoniae*, *n* = 723; *P. aeruginosa*, *n* = 325; *A. baumannii*, *n* = 73; other-EB, *n* = 610; *Staphylococcus aureus*, *n* = 294; coagulase-negative staphylococci, *n* = 216; *E. faecalis*, *n* = 223; *E. faecium, n* = 124).

The overall rate of readable zones after 4 h was 75.7% (*Enterobacterales*, 91%; *P. aeruginosa*, 0%; *A. baumannii*, 89%; *Enterococcus spp.*, 51.7%; *Staphylococcus spp*., 46.1%). After 6 h of incubation, the rate of readable zones reached 96.6% (*Enterobacterales*, 100%; *P. aeruginosa,* 89.5%; *A. baumannii*, 100%; *Enterococcus spp.*, 85.9%, *Staphylococcus spp*., 88.6%). Overall, 100% of inhibition zones were readable after 8 h and 16–20 h of incubation. Overall, the proportion of results in the the area of technical uncertainty (ATU) were 9.9%, 5.9%, 5% and 5.2% for readings at 4 h, 6 h, 8 h and 16–20 h, respectively. Categorical agreement (CA) ranged from 98.9% at 4 h to 99.4% at 16–20 h. Very major errors (VMEs) were 3.3%, 3.7% and 3.4% at 4 h, 6 h and 8 h, respectively, and decreased to 1% at 16–20 h (*p* < 0.001). The number of major errors (MEs) was low for each reading time (0.2%, 0.4% at 4 h and 6 h, respectively, and 0.3% at both 8 h and 16–20 h). No systematic difference in the distribution of discrepancies between manual and instrument-guided readings was observed.

### 2.1. E. coli and K. pneumoniae

One VIM metallo-β-lactamase-producing *E. coli*, twelve KPC-producing *K. pneumoniae* and one OXA-48-like-producing *K. pneumoniae* were identified. The ESBL phenotype was expressed by 27.2% and 33.3% of *E. coli* and *K. pneumoniae* isolates, respectively. RAST overall CA was 99.5% in both bacterial species (Table 1 and Table 2).

High numbers of readings falling under the ATU category were found for piperacillin/tazobactam in *E.coli* at 4 h, 6 h and 8 h of incubation (48.3%, 18.8% and 23%, respectively). The proportion of ATU results significantly decreased (*p* < 0.01) with readings at 16–20 h (4.1%). Rates > 10% of ATU results were also achieved for the following combinations of microorganism/drug/reading time: *E. coli*/ceftolozane/tazobactam/4 h (15.4%), *E. coli*/ceftazidime/4 h, *K. pneumoniae*/piperacillin/tazobactam/all reading times (ranging from 10.5% to 15.5%) and *K. pneumoniae*/ceftolozane/tazobactam/4 h (22.6%). Although VMEs were few for both species, rates > 2% were observed for cefotaxime (ranging from 2.2% to 6.2%), amikacin (ranging from 11.1% to 75%), tobramycin (ranging from 7.1% to 11.1%), ceftazidime (ranging from 2.3% to 7.1% in *E. coli*), piperacillin/tazobactam (4% at 4 h and 6 h in *K. pneumoniae*) and ceftolozane/tazobactam (10% at 6 h in *K. pneumoniae*).

MEs were found only for piperacillin/tazobactam in *E. coli* (2.4% at 4 h and 6 h) and *K. pneumoniae* (1.6% at 6 h). Of note, no VMEs or MEs were observed with readings at 16–20 h.

### 2.2. Enterobacterales other Than E. coli/K. pneumoniae

Bacterial species included *Enterobacter cloacae* complex (*n* = 16), *Proteus mirabilis* (*n* = 13), *Serratia marcescens* (*n* = 11), *Klebsiella oxytoca* (*n* = 10), *Klebsiella aerogenes* (*n* = 5), *Citrobacter koseri* (*n* = 2), *Klebsiella varicola* (*n* = 2), *Proteus vulgaris* (*n* = 2) and *Salmonella enterica* (*n* = 1). Five isolates expressed an ESBL phenotype (*P. mirabilis*, *n* = 4; *E. cloacae*, *n* = 1), and one isolate was a carbapenemase producer (OXA-48-like-producing *K. aerogenes*).

RAST overall CA was 98.3% (Table 3). ATU results were observed for all antibiotics, with the exception of ceftazidime/avibactam. In particular, high percentages of ATU results were observed for piperacillin/tazobactam at 4 h, 6 h and 8 h (56.5%, 22.6% and 16.1%, respectively), which decreased significantly (6.4%) with readings at 16–20 h (*p* < 0.001). High rates of VMEs for ceftazidime (22.2% at all reading times), amikacin (ranging from 28% to 42.8%) and tobramycin (ranging from 12.5% to 37.5%) were found. Of note, VMEs and MEs were obtained in ten isolates, of which six (60%) were *P. mirabilis*.

### 2.3. P. aeruginosa and A. baumannii

RAST overall CA was 99.4% and 100% for *P. aeruginosa* and *A. baumannii*, respectively (Table 4). In *P. aeruginosa*, ATU results were observed for piperacillin/tazobactam (ranging from 8.7% to 13.1%), ceftazidime/avibactam (ranging from 8.7% to 11.1%), ceftazidime (ranging from 8.1% to 17.4%), cefepime (ranging from 4.3% to 9.4%), meropenem (ranging from 4.3% to 5.1%) and levofloxacin (ranging from 4.3 to 4.7%). No VMEs and few MEs (one for ceftazidime/avibactam at 6 h, 8 h and 16–20 h, and one for ceftazidime at 16–20 h) were observed. In *A. baumannii*, high numbers of ATU results (78.6%) were observed for amikacin at 4 h. No errors were found for each reading time.

### 2.4. S. aureus and CoNS

CoNS species included *S. epidermidis* (*n* = 46), *S. hominis* (*n* = 8), *S. haemolyticus* (*n* = 7), *S. capitis* (*n* = 6), *S. simulans* (*n* = 2), *S. lugdunensis* (*n* = 2) and *S. petrasi* (*n* = 1).

CA was 98.1% and 95.9% in *S. aureus* and CoNS, respectively (Table 5). Overall ATU results were significantly higher in CoNS than *S. aureus* isolates (10% vs. 1.7%, respectively) (*p* < 0.001). The RAST test for screening of clindamycin-inducible resistance was essential to detect clindamycin-inducible resistance, allowing the identification of 42.4% (*n* = 14) and 52.3% (*n* = 23) of the total clindamycin-resistant *S. aureus* (*n* = 33) and CoNS isolates (*n* = 44), respectively. However, VMEs for clindamycin were found at 4 h, 6 h and 8 h in *S. aureus* (ranging from 6.1% to 8.3%) and CoNS isolates (ranging from 2.3% to 4.5%), whereas MEs were only observed in CoNS (range: 0–3.6%). Similarly, VMEs were observed for gentamycin at 4 h, 6 h and 8 h but not at 16–20 h. Excluding ATU results and non-readable zones, cefoxitin RAST screening identified all methicillin-resistant staphylococci at each reading time.

### 2.5. E. faecalis and E. faecium

RAST overall categorical agreement was 100% and 97.9% in *E. faecalis* and *E. faecium*, respectively (Table 6). Excluding vancomycin, for which the breakpoint of susceptibility was not available, ATU results were mainly found for linezolid in both species (3.2–11.1% in *E. faecalis*, 18.7–100% in *E. faecium*) and for gentamicin in *E. faecalis* (12.5–50%). One VME and one ME were observed in *E. faecium* for ampicillin at 4 h, and for linezolid at 6 h, respectively. A lower number of ATU results and no categorical errors were found with readings at 16–20 h.

## 3. Discussion

Although several RAST methods for BCs have been developed over time, their use has been confined to experimental studies or limited hospital settings [17,18,19,20]. Since the end of 2018, the availability of species-specific EUCAST-validated RAST breakpoints for the interpretation of inhibition zones at 4 h, 6 h and 8 h has marked a turning point in the implementation of EUCAST RAST in the BC routine of most microbiology laboratories.

In this study, EUCAST RAST showed excellent CA with automated microdilution Microscan system. However, an overall rate > 3% of false susceptibility results (VMEs) was observed, mainly due to a poor performance of the RAST for certain species/antimicrobial agent combinations. Furthermore, the limited number of isolates resistant to some of the antibiotics tested could represent an additional bias. As also shown in previous published data [13,21,22,23], VMEs were mainly observed: in aminoglycosides for *E. coli*, *K. pneumoniae* and other-EB; in cephalosporins for *E. coli* and other-EB; in gentamicin and clindamycin for *S. aureus* and CoNS. EUCAST RAST was shown to perform better for *P. aeruginosa* and *A. baumannii* (no VMEs) and for enterococci (only one false susceptible result).

Tentative breakpoints proposed for other-EB and CoNS showed satisfactory performances, with overall CA and error rates almost comparable to those obtained for *E. coli*/*K. pneumoniae* and *S. aureus*, respectively. However, high error rates were obtained for *Proteus mirabilis* isolates, suggesting that further species-specific EUCAST RAST breakpoints are desirable. In addition, a higher number of results falling in the ATU was obtained for both CoNS and other-EB. In the latter case, the explanation could be the setting of tentative breakpoints, characterized by a wider ATU aimed at minimising categorisation discrepancies.

Implementation of the RAST method for new β-lactams/β-lactamase inhibitors could be essential in geographical areas with endemic diffusion of MDR Gram-negative organisms, such as carbapenemase-resistant Enterobacterales and *P. aeruginosa* [24,25]. However, data on RAST accuracy for these antimicrobial combinations are lacking. We recently showed a good performance of EUCAST RAST in the determination of ceftazidime/avibactam susceptibility for carbapenemase-producing Enterobacterales, including both metallo-β-lactamase and serine-carbapenemase producers [16]. The present study confirmed these findings, since EUCAST RAST was demonstrated to be reliable, identifying all four ceftazidime/avibactam-resistant isolates and giving only a false resistant result for a *P. aeruginosa* isolate. However, further studies involving populations with higher rates of resistant isolates are warranted to extend these findings. To date, no study has investigated the performance of RAST in ceftolozane/tazobactam, a combination mainly used for the treatment of MDR *P. aeruginosa* and ESBL-producing Enterobacterales. In this study, 97.2–100% CA was shown for *K. pneumoniae*, *E. coli*, other-EB and *P. aeruginosa*, with no VMEs and only one ME at 6 h for a *K. pneumoniae* isolate.

Concerning the evaluation of the EUCAST RAST breakpoints for incubation at 16–20 h, we observed an improvement in performance, with higher rates of CA, reduction in the frequency of VMEs and a number of MEs almost similar to that observed at 8 h. Another advantage found with readings at 16–20 h was a strong reduction in the percentage of results falling within the ATU for piperacillin/tazobactam in *E. coli*, *K. pneumoniae* and other-EB.

De-escalation and escalation of empirical antibiotic therapy guided by RAST results is the goal for which this rapid method was developed [13,21,22,23]. However, the limited data available on this topic may, at least at present, limit the use of RAST susceptibility results to perform de-escalation interventions [13]. The delayed reading of the inhibition zones at 16–20 h, which is a very close timing to that of the standard disc-diffusion method, together with the lower rate of false susceptible results at 16–20 h compared to early readings, could strengthen the relevance of the RAST results at 16–20 h for de-escalation interventions.

## 4. Materials and Methods

The study was carried out in two tertiary-care Italian teaching hospitals (“Città della Salute e della Scienza di Torino”, Turin, and “Sant’ Orsola-Malpighi”, Bologna). During a six-month period, positive BCs were processed according to the diagnostic protocols of each laboratory (Table 7). A total of 676 positive BCs, which showed Gram-negative rods or Gram-positive cocci at Gram-staining examination, were enrolled to evaluate EUCAST RAST. EUCAST RAST was performed according to EUCAST guidelines within 2 h after the removal of a positive bottle from a BC incubator [9]. According to Gram staining and morphological classification, the predetermined antibiotic discs were placed on Mueller–Hinton plates immediately after inoculation and spreading of 150 µL of the BC fluid (Appendix A). All plates were incubated at 35 ± 1 °C in ambient air. Inhibition zones were read at 4 h, 6 h and, when possible, at 8 h and 16–20 h. The antibiotic-disc manufacturer, brand of Mueller–Hinton medium and mode of measurement of inhibition zones are reported in Table 7. According to EUCAST recommendations, the quality-control procedure was performed by both laboratories to validate the performance of antibiotic-discs, the agar used and the methods of reading inhibition zones using *E. coli* ATCC 25922, *P. aeruginosa* ATCC 27853, *S. aureus* ATCC 29213 and *E. faecalis* ATCC 29212 [26]. 

Interpretations of inhibition zone diameters were carried out in accordance with EUCAST RAST breakpoints (version 5.1) [12], and the results were compared to those obtained by reference susceptibility testing to delineate CA [27]. Polymicrobial BCs were excluded from the analysis. For other-EB, a breakpoint table was obtained by combining the RAST breakpoints for *E. coli* and *K. pneumoniae*, considering the largest values of ‘susceptible’ and ‘resistant’ (Appendix A). Since EUCAST RAST guidelines do not include breakpoints for CoNS, the RAST results were determined according to *S. aureus* RAST breakpoints. Categorical errors were classified as VME (susceptible on RAST, but resistant on reference AST), ME (resistant on RAST, but susceptible on reference AST), and minor error (mE) (susceptible or resistant on RAST, but susceptible requiring increased antibiotic exposure on reference AST). To this end, isolate/drug combinations falling within ATU or considered uninterpretable at the time of reading (i.e., insufficient growth) were excluded. Since EUCAST classifies piperacillin/tazobactam, ceftazidime, cefepime, imipenem and levofloxacin as “susceptible, but requiring increased antibiotic exposure” and not as “susceptible” for *P. aeruginosa*, discrepancies herein were considered major rather than minor errors (i.e., if resistant according to RAST but susceptible requiring increased antibiotic exposure by reference AST). The same rationale was followed for imipenem in *Morganellaceae* (*Morganella morganii*, *Proteus* spp., *Providencia* spp.) and in *Enterococcus* spp.

Comparisons involving dichotomous variables were tested using the X^2^ test. Statistical significance was set at a *p*-value < 0.05.

## 5. Conclusions

Our results showed that the RAST method is reliable for determining microbial susceptibility to main antimicrobials, including ceftazidime/avibactam and ceftolozane/tazobactam. A poorer performance was observed for certain species/antimicrobial agent combinations. The use of tentative RAST breakpoints for other-EB and CoNS has been shown to be feasible while awaiting further validated species-specific breakpoints. Readings at 16–20 h make the EUCAST RAST method applicable regardless of laboratory opening hours. Additionally, EUCAST RAST showed better performance at 16–20 h compared to early readings, giving it greater potential for antimicrobial de-escalation interventions.

## Figures and Tables

**Table 1 antibiotics-11-01404-t001:** Performance of EUCAST RAST for *E. coli* (*n* = 191).

Bacterial Species/Antibiotic	Reading Time	No. of Tests	Reference Susceptibility			Readable Zones	ATU	Categorical Agreement (%)			
			S	I	R				VME	ME	mE
Piperacillin/tazobactam	4 h	191	166	0	25	180	87	95.7	0	4	-
	6 h	191	166	0	25	191	36	97.4	0	4	-
	8 h	74	71	0	3	74	17	100	0	0	-
	16–20 h	74	71	0	3	74	3	100	0	0	-
Ceftazidime/avibactam	4 h	116	115	0	1	103	0	100	0	0	-
	6 h	116	115	0	1	116	0	100	0	0	-
	8 h	74	73	0	1	74	0	100	0	0	-
	16–20 h	74	73	0	1	74	0	100	0	0	-
Ceftolozane/tazobactam	4 h	60	59	0	1	52	8	100	0	0	-
	6 h	60	59	0	1	60	1	100	0	0	-
	8 h	60	59	0	1	60	1	100	0	0	-
	16–20 h	60	59	0	1	60	1	100	0	0	-
Cefotaxime	4 h	191	145	0	46	176	4	100	0	0	-
	6 h	191	145	0	46	191	1	99.5	1	0	-
	8 h	74	58	0	16	74	1	98.6	1	0	-
	16–20 h	74	58	0	16	74	0	100	0	0	-
Ceftazidime	4 h	191	145	2	44	177	14	99.4	1	0	0
	6 h	191	145	2	44	191	7	98.9	2	0	0
	8 h	74	58	2	14	74	3	98.6	1	0	0
	16–20 h	74	58	2	14	74	2	100	0	0	0
Imipenem	4 h	191	191	0	0	190	1	100	-	0	-
	6 h	191	191	0	0	191	0	100	-	0	-
	8 h	74	74	0	0	74	0	100	-	0	-
	16–20 h	74	74	0	0	74	0	100	-	0	-
Meropenem	4 h	191	191	0	0	185	4	100	-	0	-
	6 h	191	191	0	0	191	0	100	-	0	-
	8 h	74	74	0	0	74	0	100	-	0	-
	16–20 h	74	74	0	0	74	0	100	-	0	-
Levofloxacin	4 h	74	56	0	18	63	1	100	0	0	-
	6 h	74	56	0	18	74	0	100	0	0	-
	8 h	74	56	0	18	74	0	100	0	0	-
	16–20 h	74	56	0	18	74	3	100	0	0	-
Amikacin	4 h	191	188	-	3	188	3	98.9	2	0	-
	6 h	191	188	-	3	191	0	98.9	2	0	-
	8 h	74	73	-	1	74	0	100	0	0	-
	16–20 h	74	73	-	1	74	0	100	0	0	-
Tobramycin	4 h	74	65	-	9	72	1	100	0	0	-
	6 h	74	65	-	9	74	2	98.6	1	0	-
	8 h	74	65	-	9	74	2	98.6	1	0	-
	16–20 h	74	65	-	9	74	1	100	0	0	-
All	4 h	1470	1321	2	147	1386	123	99.4	3	4	0
	6 h	1470	1321	2	147	1470	47	99.3	6	4	0
	8 h	726	661	2	63	726	24	99.6	3	0	0
	16–20 h	726	661	2	63	726	10	100	0	0	0

Abbreviations: S, susceptible; I, susceptible at increased exposure; R, resistant; ATU, area of technical uncertainty; VME, very major error; ME, major error; mE, minor error.

**Table 2 antibiotics-11-01404-t002:** Performance of EUCAST RAST for *K. pneumoniae* (*n* = 87).

Bacterial Species/Antibiotic	Reading Time	No. of Tests	Reference Susceptibility			Readable Zones	ATU	Categorical Agreement (%)			
			S	I	R				VME	ME	mE
Piperacillin/tazobactam	4 h	87	62	-	25	84	13	98.6	1	0	-
	6 h	87	62	-	25	87	10	97.4	1	1	-
	8 h	38	26	-	12	38	7	100	0	0	-
	16–20 h	38	26	-	12	38	4	100	0	0	-
Ceftazidime/avibactam	4 h	87	84	-	3	83	0	100	0	0	-
	6 h	87	84	-	3	87	0	100	0	0	-
	8 h	38	35	-	3	38	0	100	0	0	-
	16–20 h	38	35	-	3	38	0	100	0	0	-
Ceftolozane/tazobactam	4 h	38	28	-	10	31	7	100	0	0	-
	6 h	38	28	-	10	38	2	97.2	1	0	-
	8 h	38	28	-	10	38	3	100	0	0	-
	16–20 h	38	28	-	10	38	5	100	0	0	-
Cefotaxime	4 h	87	48	0	39	84	2	100	0	0	-
	6 h	87	48	0	39	87	4	98.8	1	0	-
	8 h	38	19	0	19	38	1	100	0	0	-
	16–20 h	38	19	0	19	38	0	100	0	0	-
Ceftazidime	4 h	87	48	0	39	83	4	100	0	0	-
	6 h	87	48	0	39	87	2	100	0	0	-
	8 h	38	20	0	18	38	1	100	0	0	-
	16–20 h	38	20	0	18	38	1	100	0	0	-
Imipenem	4 h	87	78	0	9	85	1	100	0	0	-
	6 h	87	78	0	9	87	1	100	0	0	-
	8 h	38	32	0	6	38	0	100	0	0	-
	16–20 h	38	32	0	6	38	0	100	0	0	-
Meropenem	4 h	87	76	1	10	85	5	100	0	0	0
	6 h	87	76	1	10	87	4	100	0	0	0
	8 h	38	31	1	6	38	0	100	0	0	0
	16–20 h	38	31	1	6	38	0	100	0	0	0
Levofloxacin	4 h	38	22	0	16	36	0	100	0	0	-
	6 h	38	22	0	16	38	0	100	0	0	-
	8 h	38	22	0	16	38	0	100	0	0	-
	16–20 h	38	22	0	16	38	0	100	0	0	-
Amikacin	4 h	87	78	-	9	85	2	98.8	1	0	-
	6 h	87	78	-	9	87	0	97.7	2	0	-
	8 h	38	30	-	8	38	0	94.7	2	0	-
	16–20 h	38	30	-	8	38	0	100	0	0	-
Tobramycin	4 h	38	24	-	14	36	1	97.1	1	0	-
	6 h	38	24	-	14	38	0	100	0	0	-
	8 h	38	24	-	14	38	0	100	0	0	-
	16–20 h	38	24	-	14	38	0	100	0	0	-
All	4 h	723	548	1	174	692	35	99.5	3	0	0
	6 h	723	548	1	174	723	23	99.1	5	1	0
	8 h	380	267	1	112	380	12	99.4	2	0	0
	16–20 h	380	267	1	112	380	10	100	0	0	0

Abbreviations: S, susceptible; I, susceptible at increased exposure; R, resistant; ATU, area of technical uncertainty; VME, very major error; ME, major error; mE, minor error.

**Table 3 antibiotics-11-01404-t003:** Performance of EUCAST RAST for Enterobacterales species other than *E.coli*/*K. pneumoniae* (*n* = 61).

Bacterial Species/Antibiotic	Reading Time	No. of Tests	Reference Susceptibility			Readable Zones	ATU	Categorical Agreement (%)			
			S	I	R				VME	ME	mE
Piperacillin/tazobactam	4 h	61	53	0	8	45	26	100	0	0	0
	6 h	61	53	-	8	61	14	100	0	0	0
	8 h	61	53	-	8	61	10	100	0	0	0
	16–20 h	61	53	-	8	61	4	100	0	0	0
Ceftazidime/avibactam	4 h	61	61	-	0	43	0	100	0	0	0
	6 h	61	61	-	0	61	0	100	0	0	0
	8 h	61	61	-	0	61	0	100	0	0	0
	16–20 h	61	61	-	0	61	0	100	0	0	0
Ceftolozane/tazobactam	4 h	61	59	-	2	44	2	100	0	0	0
	6 h	61	59	-	2	61	4	100	0	0	0
	8 h	61	59	-	2	61	3	100	0	0	0
	16–20 h	61	59	-	2	61	4	100	0	0	0
Cefotaxime	4 h	61	48	0	13	40	3	97.3	1	0	-
	6 h	61	48	0	13	61	5	100	0	0	-
	8 h	61	48	0	13	61	3	100	0	0	-
	16–20 h	61	48	0	13	61	1	100	0	0	-
Ceftazidime	4 h	61	52	1	8	40	1	92.3	2	0	1
	6 h	61	52	1	8	61	0	95.1	2	0	1
	8 h	61	52	1	8	61	0	95.1	2	0	1
	16–20 h	61	52	1	8	61	2	95.1	2	0	1
Imipenem	4 h	61	45	15	1	50	2	100	0	0	0
	6 h	61	45	15	1	61	8	100	0	0	0
	8 h	61	45	15	1	61	7	98.1	0	1	0
	16–20 h	61	45	15	1	61	6	100	0	0	0
Meropenem	4 h	61	60	0	1	50	8	100	0	0	-
	6 h	61	60	0	1	61	0	100	0	0	-
	8 h	61	60	0	1	61	0	100	0	0	-
	16–20 h	61	60	0	1	61	0	100	0	0	-
Levofloxacin	4 h	61	54	0	7	51	3	100	0	0	-
	6 h	61	54	0	7	61	1	98.3	0	1	-
	8 h	61	54	0	7	61	1	98.3	0	1	-
	16–20 h	61	54	0	7	61	8	98.1	0	1	-
Amikacin	4 h	61	54	-	7	56	2	94.4	3	0	-
	6 h	61	54	-	7	61	4	96.5	2	0	-
	8 h	61	54	-	7	61	4	96.5	2	0	-
	16–20 h	61	54	-	7	61	2	98.3	1	0	-
Tobramycin	4 h	61	54	-	7	56	2	94.4	3	0	-
	6 h	61	54	-	7	61	4	96.5	2	0	-
	8 h	61	54	-	7	61	4	96.5	2	0	-
	16–20 h	61	54	-	7	61	2	98.3	1	0	-
All	4 h	610	540	16	54	475	49	97.6	9	0	1
	6 h	610	540	16	54	610	40	98.6	6	1	1
	8 h	610	540	16	54	610	32	98.4	6	2	1
	16–20 h	610	540	16	54	610	29	99	4	1	1

Abbreviations: S, susceptible; I, susceptible at increased exposure; R, resistant; ATU, area of technical uncertainty; VME, very major error; ME, major error; mE, minor error.

**Table 4 antibiotics-11-01404-t004:** Performance of EUCAST RAST for *P. aeruginosa* (*n* = 42) and *A. baumannii* (*n* = 15).

Bacterial Species/Antimicrobial	Reading Time	No. of Isolates Tested	Reference Susceptibility			Readable Zones	ATU	Categorical Agreement (%)			
			S	I	R				VME	ME	mE
*P. aeruginosa* (*n* = 42)											
Piperacillin/tazobactam	6 h	42	-	39	3	38	5	100	0	0	-
	8 h	23	-	23	0	23	2	100	-	0	-
	16–20 h	23	-	23	0	23	2	100	-	0	-
Ceftazidime/avibactam	6 h	23	23	-	0	18	2	93.7	-	1	-
	8 h	23	23	-	0	23	2	100	-	1	-
	16–20 h	23	23	-	0	23	2	95.2	-	1	-
Ceftolozane/tazobactam	6 h	23	23	-	0	22	0	100	-	0	-
	8 h	23	23	-	0	23	0	100	-	0	-
	16–20 h	23	23	-	0	23	0	100	-	0	-
Ceftazidime	6 h	42	-	37	5	37	3	100	0	0	-
	8 h	23	-	22	1	23	4	100	0	0	-
	16–20 h	23	-	22	1	23	0	95.6	0	1	-
Cefepime	6 h	42	-	37	5	32	3	100	0	0	-
	8 h	23	-	21	2	23	0	100	0	0	-
	16–20 h	23	-	21	2	23	1	100	0	0	-
Imipenem	6 h	23	-	17	6	21	0	100	0	0	-
	8 h	23	-	17	6	23	0	100	0	0	-
	16–20 h	23	-	17	6	23	0	100	0	0	-
Meropenem	6 h	42	35	1	6	39	2	100	0	0	0
	8 h	23	18	1	4	23	1	100	0	0	0
	16–20 h	23	18	1	4	23	1	95.4	0	0	1
Levofloxacin	6 h	23	-	21	2	21	1	100	0	0	-
	8 h	23	-	21	2	23	0	100	0	0	-
	16–20 h	23	-	21	2	23	1	100	0	0	-
Amikacin	6 h	42	42	-	0	42	0	100	-	0	-
	8 h	23	23	-	0	23	0	100	-	0	-
	16–20 h	23	23	-	0	23	0	100	-	0	-
Tobramycin	6 h	23	22	-	1	21	0	100	0	0	-
	8 h	23	22	-	1	23	0	100	0	0	-
	16–20 h	23	22	-	1	23	0	100	0	0	-
All	6 h	325	145	152	28	291	16	99.6	0	1	0
	8 h	230	106	105	16	230	9	99.5	0	1	0
	16–20 h	230	106	105	16	230	7	98.6	0	2	1
*A. baumannii* (*n* = 15)											
Imipenem	4 h	15	3	0	12	13	0	100	0	0	0
	6 h	15	3	0	12	15	0	100	0	0	0
	8 h	13	3	0	10	13	0	100	0	0	-
Meropenem	4 h	15	3	0	12	13	1	100	0	0	-
	6 h	15	3	0	12	15	0	100	0	0	-
	8 h	13	3	0	10	13	0	100	0	0	-
Levofloxacin	4 h	15	3	0	12	14	0	100	0	0	-
	6 h	15	3	0	12	15	0	100	0	0	-
	8 h	13	3	0	10	13	0	100	0	0	-
Amikacin	4 h	15	5	0	10	14	11	100	0	0	-
	6 h	15	5	0	10	15	2	100	0	0	-
	8 h	13	4	0	9	13	0	100	0	0	-
Tobramycin	4 h	13	3	0	10	11	1	100	0	0	-
	6 h	13	3	0	10	13	2	100	0	0	-
	8 h	13	3	0	10	13	0	100	0	0	-
All	4 h	73	17	0	56	65	13	100	0	0	0
	6 h	73	17	0	56	73	4	100	0	0	0
	8 h	65	16	0	49	65	0	100	0	0	0

Abbreviations: S, susceptible; I, susceptible at increased exposure; R, resistant; ATU, area of technical uncertainty; VME, very major error; ME, major error; mE, minor error.

**Table 5 antibiotics-11-01404-t005:** Performance of EUCAST RAST for *S. aureus* (*n* = 98) and CoNS (*n* = 72).

Bacterial Species/Antimicrobial	Reading Time	No. of Isolates Tested	Reference Susceptibility			Readable Zones	ATU	Categorical Agreement (%)			
			S	I	R				VME	ME	mE
*S. aureus* (*n* = 98)											
Cefoxitin	4 h	98	73	0	25	74	0	98.6	0	1	-
	6 h	98	73	0	25	98	0	99	0	1	-
	8 h	37	25	0	12	37	1	100	0	0	-
	16–20 h	37	25	0	12	37	0	97.3	0	1	-
Clindamycin	4 h	98	65	0	33	60	2	96.5	2	0	0
	6 h	98	65	0	33	98	1	97.9	2	0	0
	8 h	37	25	0	12	37	2	97.1	1	0	0
	16–20 h	37	25	0	12	37	3	100	0	0	0
Gentamicin	4 h	98	90	0	8	63	1	96.8	2	0	-
	6 h	98	90	0	8	98	2	97.9	2	0	-
	8 h	37	34	0	3	37	0	97.3	1	0	-
	16–20 h	37	34	0	3	37	0	100	0	0	-
All	4 h	294	228	0	66	197	3	97.4	4	1	0
	6 h	294	228	0	66	294	3	98.3	4	1	0
	8 h	111	84	0	27	111	3	98.1	2	0	0
	16–20 h	111	84	0	27	111	3	99.1	0	1	0
*CONS* (*n* = 72)											
Cefoxitin	4 h	72	21	-	51	14	2	83.3	0	2	-
	6 h	72	21	-	51	54	2	98.1	0	1	-
	8 h	72	21	-	51	72	2	98.6	0	1	0
	16–20 h	72	21	-	51	72	1	98.6	0	2	0
Clindamycin	4 h	72	28	0	44	12	2	90	1	0	-
	6 h	72	28	0	44	52	6	93.5	2	1	-
	8 h	72	28	0	44	72	14	94.8	2	1	-
	16–20 h	72	28	0	44	72	6	98.5	0	1	-
Gentamicin	4 h	72	28	0	44	12	2	80	2	0	0
	6 h	72	28	0	44	52	6	91.3	3	1	0
	8 h	72	28	0	44	72	14	94.8	2	1	0
	16–20 h	72	28	0	44	72	6	98.5	0	1	0
All	4 h	216	77	0	139	38	6	84.4	3	2	0
	6 h	216	77	0	139	158	14	94.4	5	3	0
	8 h	216	77	0	139	216	30	96.2	4	3	0
	16–20 h	216	77	0	139	216	13	98.5	0	3	0

Abbreviations: S, susceptible; I, susceptible at increased exposure; R, resistant; ATU, area of technical uncertainty; VME, very major error; ME, major error; mE, minor error.

**Table 6 antibiotics-11-01404-t006:** Performance of EUCAST RAST for *E. faecalis* (*n* = 48) and *E. faecium* (*n* = 27).

Bacterial Species/Antimicrobial	Reading Time	No. of Isolates Tested	Reference Susceptibility			Readable Zones	ATU	Categorical Agreement (%)			
			S	I	R				VME	ME	mE
*E. faecalis* (*n* = 48)											
Ampicillin	4 h	48	46	0	2	37	0	100	0	0	-
	6 h	48	46	0	2	48	0	100	0	0	-
	8 h	31	30	0	1	31	0	100	0	0	-
Imipenem	4 h	48	-	46	2	37	0	100	0	0	-
	6 h	48	-	46	2	48	0	100	0	0	-
	8 h	31	-	30	1	31	0	100	0	0	-
Vancomycin	4 h	48	46	46	2	39	37	100	-	0	-
	6 h	48	46	46	2	48	46	100	-	0	-
	8 h	31	29	29	2	31	29	100	-	0	-
Linezolid	4 h	48	48	0	0	27	3	100	-	0	-
	6 h	48	48	0	0	42	4	100	-	0	-
	8 h	31	31	0	0	31	1	100	-	0	-
Gentamicin	4 h	31	27	0	4	14	7	100	0	0	0
	6 h	31	27	0	4	25	3	100	0	0	0
	8 h	31	27	0	4	31	0	100	0	0	0
All	4 h	223	167	46	10	154	47	100	0	0	0
	6 h	223	167	46	10	211	53	100	0	0	0
	8 h	155	117	30	8	155	30	100	0	0	0
*E. faecium* (*n* = 27)											
Ampicillin	4 h	27	4	0	23	12	0	91.7	1	0	0
	6 h	27	4	0	23	16	0	93.7	1	0	0
	8 h	16	2	0	14	16	0	100	0	0	0
Imipenem	4 h	27	-	5	22	13	3	100	0	0	0
	6 h	27	-	5	22	16	3	100	0	0	0
	8 h	16	-	3	13	16	0	100	0	0	0
Vancomycin	4 h	27	22	0	5	18	17	100	0	0	0
	6 h	27	22	0	5	26	23	100	0	0	0
	8 h	16	12	0	4	16	14	100	0	0	0
Linezolid	4 h	27	27	0	0	11	11	-	-	-	-
	6 h	27	27	0	0	17	4	92.3	0	1	0
	8 h	16	16	0	0	16	3	100	0	0	0
Gentamicin	4 h	16	12	0	4	0	-	-	-	-	-
	6 h	16	12	0	4	12	0	100	0	0	0
	8 h	16	12	0	4	16	0	100	0	0	0
All	4 h	124	65	5	54	54	31	95.6	1	0	0
	6 h	124	65	5	54	87	30	96.5	1	1	0
	8 h	80	42	3	35	80	17	100	0	0	0

Abbreviations: S, susceptible; I, susceptible at increased exposure; R, resistant; ATU, area of technical uncertainty; VME, very major error; ME, major error; mE, minor error.

**Table 7 antibiotics-11-01404-t007:** Blood-culture systems, species-identification methods, antimicrobial susceptibility testing methods, carbapenemase and ESBL detection methods, Mueller–Hinton media, antibiotic-discs manufacturers and inhibition-zone reading methods used in the two clinical microbiology laboratories.

						EUCAST RAST		
	BC System	Identification System	AST	Carbapenemase Detection Method	ESBL Detection Method	Mueller–Hinton Manufacturer	Discs Manufacturer	Reading Method
Turin	BactAlert Virtuo (Biomerieux, Marcy l’ Etoile, France)	MALDI-TOF MS (Bruker, Rosenheim, Germany)	Microscan Panels (Beckman Coulter, Beverly, MA, USA)	Lateral flow immunoassay (NG Biotech, Guipry, France) or CARBA R molecular testing (Cepheid, Sunnyvale, CA, USA)	Microscan confirmatory testing and lateral flow immunoassay (NG Biotech)	Becton Dickinson GmbH	Oxoid Ltd., Basingstoke, UK	Manually using a calliper
Bologna	BACTEC FX (Becton Dickinson, Franklin Lakes, NJ, USA)	MALDI-TOF MS (Bruker)	Microscan Panels (Beckman Coulter)	Lateral flow immunoassay (NG Biotech) or CARBA R molecular testing (Cepheid)	Microscan confirmatory testing	Becton Dickinson GmbH	Oxoid Ltd.	BIOMIC V3 (Giles Scientific Inc., Santa Barbara, CA, USA)

## Data Availability

The authors confirm that the data supporting the findings of this study are available from the corresponding author on reasonable request.

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
