# Peer review of "Multicentre Evaluation of the EUCAST Rapid Antimicrobial Susceptibility Testing (RAST) Extending Analysis to 16–20 Hours Reading Time"

_antibiotics, 2022, doi:10.3390/antibiotics11101404_

Round 1
Reviewer 1 Report
This paper by Bianco et al. describes a study where the EUCAST RAST has been evaluated and compared to Microscan AST-testing. Overall, the data are interesting and relevant and the methodology sound. The authors also present data on other species than those validated by the EUCAST and most importantly also data for overnight reading (16-20h) as this is a practical issue in the lab and increases the usability of the EUCAST RAST method. Minor comments: Tables: The numbering of the tables is incorrect (table 1 and table 4 are inverted).Tables could be made smaller to cover fewer pages and to facilitate the reading of results. Line 20: “to evaluate EUCAST RAST” can be deleted Lines 45-46: The sentence is not clear. Please rephrase Line 118: Replace E. Coli with E. coli Line 199: Add ATU% for gentamicin in E. faecalis Lines 235-236: The sentence is not clear. Please rephraseAuthor Response
We would like to thank the Editorial Team for his helpful suggestions, which in our view have enhanced the quality and strength of our study. We hope that in this revised version the manuscript is now suitable for publication in Antibiotics.
Please, note that the changes to the original manuscript have been highlighted in the text. The response to the Reviewer’s comments and ensuing modifications in the manuscript are also clearly indicated.
Comments from Reviewer#1 and point-by-point answers
This paper by Bianco et al. describes a study where the EUCAST RAST has been evaluated and compared to Microscan AST-testing. Overall, the data are interesting and relevant and the methodology sound. The authors also present data on other species than those validated by the EUCAST and most importantly also data for overnight reading (16-20h) as this is a practical issue in the lab and increases the usability of the EUCAST RAST method. Minor comments:
1) Tables: The numbering of the tables is incorrect (table 1 and table 4 are inverted). Tables could be made smaller to cover fewer pages and to facilitate the reading of results.
We thank the Reviewer for this comment. Accordingly, the order of the tables has been corrected. In addition, the number of tables has been increased to make it easier for readers to read the results.
2) Line 20: “to evaluate EUCAST RAST” can be deleted
We thank the Reviewer for this comment. Accordingly, we corrected the sentence.
3) Lines 45-46: The sentence is not clear.
We thank the Reviewer for this comment. Accordingly, we rephrased the sentence.
4) Please rephrase Line 118: Replace E. Coli with E. coli
We thank the Reviewer for this comment. Accordingly, we removed this mistake.
5) Line 199: Add ATU% for gentamicin in E. faecalis
We thank the Reviewer for this comment. Accordingly, we added ATU% values.
6) Lines 235-236: The sentence is not clear. Please rephrase
We thank the Reviewer for this comment. Accordingly, we rephrased the sentence.
Reviewer 2 Report
Dear Authors,
Your presented manuscript provides important informations for using new antibiotic combinations in the RAST method.
However, the present version of the manuscript contains a number of technical faults that need to be resolved. All are presented in the PDF version of the revision. The tables are not the ones that are related to the text. Also, tables need to be better presented, with more text wrapping or by dividing to separate ones for respected bacteria.
Also, the number of tested isolates in different reading times is not the same. Line 82 - Reading of inhibition zones is done on 4 and 6h, but on 8 and 16-20h when possible? On the other hand, in line 124, - Overall, 100% of inhibition zones were readable after 8h and 16-20h of inhibition. Please explain.
In tables with susceptibility data, in many cases, we have different numbers of tested isolates in different readable times for the same antibiotic and species. Please explain.
Methods could be more closely explained. At least, provide a flowchart of the methods that are presented in Table 4. It would be more clearly presented for readers who are not clinical medical microbiologists, what was done for each BC/sample.
Author Response
We would like to thank the Editorial Team for his helpful suggestions, which in our view have enhanced the quality and strength of our study. We hope that in this revised version the manuscript is now suitable for publication in Antibiotics.
Please, note that the changes to the original manuscript have been highlighted in the text. The response to the Reviewer’s comments and ensuing modifications in the manuscript are also clearly indicated.
Comments from Reviewer#2 and point-by-point answers:
Dear Authors,
Your presented manuscript provides important informations for using new antibiotic combinations in the RAST method.
However, the present version of the manuscript contains a number of technical faults that need to be resolved. All are presented in the PDF version of the revision.
1) The tables are not the ones that are related to the text. Also, tables need to be better presented, with more text wrapping or by dividing to separate ones for respected bacteria.
We thank the Reviewer for this comment. Accordingly, the order of the tables has been corrected. In addition, the number of tables has been increased to make it easier for readers to read the results.
2) Also, the number of tested isolates in different reading times is not the same. Line 82 - Reading of inhibition zones is done on 4 and 6h, but on 8 and 16-20h when possible?
We thank the Reviewer for this comment. The readings at 8h and 16-20h were not taken for all isolates. As stated in "materials and methods", the 8h and 16-20h readings were carried out "when possible", compatible with the opening hours of the laboratory.
3) On the other hand, in line 124, - Overall, 100% of inhibition zones were readable after 8h and 16-20h of inhibition. Please explain.
We thank the Reviewer for this comment. The sentence is correct: all inhibition zones were readable at 8h and 16-20h.
4) In tables with susceptibility data, in many cases, we have different numbers of tested isolates in different readable times for the same antibiotic and species. Please explain.
We thank the Reviewer for this comment. The readings at 8h and 16-20h were not taken for all isolates. As stated in "materials and methods", the 8h and 16-20h readings were carried out "when possible", compatible with the opening hours of the laboratory.
5) Methods could be more closely explained. At least, provide a flowchart of the methods that are presented in Table 4. It would be more clearly presented for readers who are not clinical medical microbiologists, what was done for each BC/sample.
We thank the Reviewer for this comment. The purpose of the study is to evaluate the EUCAST RAST method, not to propose a workflow in the blood culture routine. In fact, a limited number of positive blood cultures were selected from the two laboratories during the study period. The purpose of table 7 is to indicate the materials used for the reference IDs and ASTs, and for the materials used for the RAST method. Therefore, the addition of a flowchart would be misleading for the purposes of the study.
Reviewer 3 Report
The main question addressed by this research is how well certain extending the reading time can improve EUCAST RAST results. The topic is original as there are no other similar studies in the literature for the particular extended time. This study will help physicians to be able to identify the suitable antimicrobials more accurately for curing infections more effectively. The methodology was sound and well-designed - no corrections. The conclusions are consistent with the evidence and arguments presented and the main question posed is well addressed. The references are appropriate and up-to-date. No additional comments on the tables and figures.
Minor corrections:
1. The title is confusing, especially the second half.
2. Abstract: You need to start with some background about EUCAST RAST
3. Abstract: species related to what?
4. Abstract: define all abbreviations: BC, VMA, CA etc.
5. Introduction: Reference 16 is in wrong format.
6. Introduction: Clarify aims and objectives, this paragraph is confusing
7. Tables: Ensure table titles are on the same page with the tables.
8. Line 236, reference 16 is in the wrong format
Author Response
We would like to thank the Editorial Team for his helpful suggestions, which in our view have enhanced the quality and strength of our study. We hope that in this revised version the manuscript is now suitable for publication in Antibiotics.
Please, note that the changes to the original manuscript have been highlighted in the text. The response to the Reviewer’s comments and ensuing modifications in the manuscript are also clearly indicated.
Comments from Reviewer#3 and point-by-point answers:
The main question addressed by this research is how well certain extending the reading time can improve EUCAST RAST results. The topic is original as there are no other similar studies in the literature for the particular extended time. This study will help physicians to be able to identify the suitable antimicrobials more accurately for curing infections more effectively. The methodology was sound and well-designed - no corrections. The conclusions are consistent with the evidence and arguments presented and the main question posed is well addressed. The references are appropriate and up-to-date. No additional comments on the tables and figures.
Minor corrections:
1) The title is confusing, especially the second half.
We thank the Reviewer for this comment. Accordingly we removed the second half of the title.
2) Abstract: You need to start with some background about EUCAST RAST
The maximum limit for the abstract is 250 words.
The study data to be presented are numerous and the EUCAST method is widely described and known (EUCAST website and literature works). We therefore preferred to start with the scope of the work.
3) Abstract: species related to what?
We thank the Reviewer for this comment. Accordingly we modified the text.
4) Abstract: define all abbreviations: BC, VMA, CA etc.
We thank the Reviewer for this comment. Accordingly we modified the text.
5) Introduction: Reference 16 is in wrong format.
We thank the Reviewer for this comment. Accordingly we modified the format.
6) Introduction: Clarify aims and objectives, this paragraph is confusing
We thank the Reviewer for this comment. Accordingly we modified the text.
7) Tables: Ensure table titles are on the same page with the tables.
We thank the Reviewer for this comment. Accordingly we modified the text.
8) Line 236, reference 16 is in the wrong format
We thank the Reviewer for this comment. Accordingly we modified the format.